# A Fully Human IgE Specific for CD38 as a Potential Therapy for Multiple Myeloma

**DOI:** 10.3390/cancers15184533

**Published:** 2023-09-13

**Authors:** Pierre V. Candelaria, Miguel Nava, Tracy R. Daniels-Wells, Manuel L. Penichet

**Affiliations:** 1Division of Surgical Oncology, Department of Surgery, David Geffen School of Medicine, University of California, Los Angeles (UCLA), Los Angeles, CA 90095, USA; 2Department of Microbiology, Immunology, and Molecular Genetics, David Geffen School of Medicine, University of California, Los Angeles (UCLA), Los Angeles, CA 90095, USA; 3UCLA AIDS Institute, Los Angeles, CA 90095, USA; 4UCLA Jonsson Comprehensive Cancer Center, Los Angeles, CA 90095, USA; 5The Molecular Biology Institute, University of California, Los Angeles (UCLA), Los Angeles, CA 90095, USA

**Keywords:** CD38, IgE, immunotherapy, AllergoOncology, multiple myeloma

## Abstract

**Simple Summary:**

Multiple myeloma (MM) is blood cancer of plasma cells. Plasma cells are white blood cells, which are part of the immune system. Malignant plasma cells are found in the bone marrow and are difficult to treat. There have been many new therapies developed in recent years, but the disease is still considered incurable. Some of the treatments for MM are antibodies of the IgG class. We have developed a fully human antibody of another antibody class, IgE, targeting the CD38 molecule that is expressed on the surface of several cancers including MM. Our goal is to use the anti-CD38 IgE antibody to recruit different types of immune cells to kill malignant cells. In this article, we report that the antibody shows anti-cancer effects against MM cells. Thus, this IgE antibody should be further explored as a novel treatment for MM.

**Abstract:**

Multiple myeloma (MM) is an incurable malignancy of plasma cells and the second most common hematologic malignancy in the United States. Although antibodies in clinical cancer therapy are generally of the IgG class, antibodies of the IgE class have attractive properties as cancer therapeutics, such as their high affinity for Fc receptors (FcεRs), the low serum levels of endogenous IgE allowing for less competition for FcR occupancy, and the lack of inhibitory FcRs. Importantly, the FcεRs are expressed on immune cells that elicit antibody-dependent cell-mediated cytotoxicity (ADCC), antibody-dependent cell-mediated phagocytosis (ADCP), and/or antigen presentation such as mast cells, eosinophils, macrophages, and dendritic cells. We now report the development of a fully human IgE targeting human CD38 as a potential MM therapy. We targeted CD38 given its high and uniform expression on MM cells. The novel anti-CD38 IgE, expressed in mammalian cells, is properly assembled and secreted, exhibits the correct molecular weight, binds antigen and the high affinity FcεRI, and induces degranulation of FcεRI expressing cells in vitro and also in vivo in transgenic BALB/c mice expressing human FcεRIα. Moreover, the anti-CD38 IgE induces ADCC and ADCP mediated by monocytes/macrophages against human MM cells (MM.1S). Importantly, the anti-CD38 IgE also prolongs survival in a preclinical disseminated xenograft mouse model using SCID-Beige mice and human MM.1S cells when administered with human peripheral blood mononuclear cells (PBMCs) as a source of monocyte effector cells. Our results suggest that anti-CD38 IgE may be effective in humans bearing MM and other malignancies expressing CD38.

## 1. Introduction

Multiple myeloma (MM) is a malignancy of plasma cells characterized by osteolytic bone lesions, hypercalcemia, renal failure, anemia, and neuropathy [1,2]. It is the second most common hematologic malignancy in the United States and remains incurable despite the advances in treatment strategies including immunomodulatory drugs, proteasome inhibitors, and monoclonal antibodies of the IgG class [1,2,3,4]. One such monoclonal antibody is daratumumab (Darzalex^®^), which is a first-in-class anti-CD38 monoclonal antibody for MM treatment approved by the Food and Drug Administration (FDA) as a monotherapy for relapsed and refractory MM in 2015 and in combination with either bortezomib and dexamethasone or lenalidomide and dexamethasone in 2016 [2,5]. Daratumumab, a fully human IgG1 antibody, eliminates MM cells expressing CD38 through various direct mechanisms: antibody-dependent cell-mediated cytotoxicity (ADCC), antibody-dependent cell mediated-phagocytosis (ADCP), complement-dependent cytotoxicity (CDC), and induction of programmed cell death (apoptosis) in the presence of a cross-linking antibody [6,7,8] as well as immunomodulatory mechanisms [9,10]. Since then, another anti-CD38 monoclonal antibody (humanized IgG1), isatuximab (Sarclisa^®^), which binds a distinct epitope and has shown to mediate MM cell death via ADCC, ADCP, CDC and the direct induction of caspase-dependent apoptosis without the need for a cross-linking antibody [11,12], was approved by the FDA in 2020, in combination with pomalidomide and dexamethasone, as a therapy for MM patients who have had at least two prior therapies [11,12].

Multiple epidemiological studies on the link between allergies and cancer suggest a lower risk of certain cancers among those with a history of allergies [13,14,15]. Interestingly, high levels of total plasma IgE have been linked with low risk of chronic lymphocytic leukemia (CLL) and possibly of MM [16], and higher levels of polyclonal IgE in non-allergic individuals are correlated with lower disease incidence and higher survival in MM [17]. In addition, individuals with IgE deficiency have higher frequency of malignancies, including MM [18]. Importantly, IgE antibodies isolated from pancreatic cancer patients mediate ADCC against cancer cells [19]. Taken together, these results suggest a potential natural protective role of IgE against cancer.

In general, antibodies for cancer therapy in the clinic are of the IgG class [4,20]. However, antibodies of the IgE class, well known to be part of allergic immune responses, also have attractive properties as cancer therapeutics [13,14,21,22], as explained hereunder. There are two FcεRs: FcεRI that binds IgE with high affinity (*K*a = 10^10^ M^−1^) and is expressed on human mast cells, monocytes, macrophages, eosinophils, basophils, Langerhans cells, and dendritic cells (DC); and the FcεRII (CD23) that in its trimeric form binds IgE with lower, but still high affinity (*K*a = 10^8^ M^−1^), expressed on human eosinophils, monocytes, macrophages, and DC. Importantly, the aforementioned cells expressing FcεRs are capable of eliciting ADCC, ADCP, and/or antigen presentation. Thus, IgE binds to FcεRs with extremely high affinity, which in the case of FcεRI is two–three orders of magnitude higher than that of IgG for the FcγRs (FcγRI-III). In fact, IgE is considered a cytophilic antibody with a long half-life on the surface of effector cells. This property would thereby allow effector cells armed with IgE to penetrate the tumor microenvironment with subsequent antitumor activity. Moreover, the low serum levels of endogenous IgE in circulation, only 0.02% compared to 85% of IgG, allow for less competition for FcR occupancy. In fact, the loss of tumor-specific IgG from effector cells due to the high levels of competing serum IgG for host FcRs limits ADCC. Additionally, IgE does not have an inhibitory FcR, while IgG binds to the inhibitory FcγIIB, decreasing Fc-dependent effector functions such as ADCC.

Our group and others have developed IgE antibodies targeting different solid tumor antigens, such as HER2/*neu*, prostate-specific antigen (PSA), folate receptor alpha (FRα), and MUC1, that exhibit antitumor activity and are well-tolerated in animal models, including non-human primates [23,24,25,26,27,28,29]. Importantly, the anti-FRα IgE (MOv18 IgE), a mouse/human chimeric antibody, was the first antibody of IgE class to enter a Phase I clinical trial in patients with advanced solid tumors (NCT02546921), in which it was generally well-tolerated and showed preliminary evidence for antitumor efficacy [15,30,31]. Research on cancer and IgE pertains to the novel field of AllergoOncology, which aims to reveal the function of IgE-mediated immune responses against cancer and to develop IgE-based therapies against malignant diseases [13,32].

Given the relevance of CD38 as a MM antigen and the attractive properties of the IgE antibody, we developed a fully human anti-CD38 IgE and evaluated its properties, including anti-cancer effects, in vitro and in vivo.

## 2. Materials and Methods

### 2.1. Cell Lines

Human MM.1S myeloma cells of African American origin (ATCC^®^ CRL-2974™) and murine Sp2/0-Ag14 myeloma cells (ATCC^®^ CRL-1581™) were purchased from the American Type Culture Collection (ATCC, Manassas, VA, USA). RBL SX-38, a rat basophilic leukemia cell line that expresses the human α, β, and γ chains of FcεRI (complete human receptor) [33], was kindly provided by Dr. Jean-Pierre Kinet (Beth Israel Deaconess Medical Center, Boston, MA, USA). MM.1S cells were cultured in RPMI 1640 (Thermo Fisher Scientific, Waltham, MA, USA) supplemented with 10% heat-inactivated fetal bovine serum (FBS, Atlanta Biologicals, Atlanta, GA, USA) and penicillin/streptomycin (Thermo Fisher Scientific). Sp2/0 Ag14 cells were cultured in IMDM supplemented with 5% heat-inactivated GemCell™ newborn calf serum (Gemini Bio-Products, Sacramento, CA, USA) and penicillin/streptomycin (Thermo Fisher Scientific). RBL SX-38 cells were cultured in IMDM supplemented with 10% heat-inactivated FBS, 1 mg/mL G418 (an aminoglycoside antibiotic also known as “Geneticin”, Thermo Fisher Scientific), and penicillin/streptomycin. All cell cultures were incubated at 5% CO_2_ and 37 °C.

### 2.2. Antibody Development

The fully human anti-CD38 IgE and anti-CD38 IgG1 antibodies were constructed using synthetic DNA (ATUM, Newark, CA, USA) encoding the variable light chain (V_L_) and the variable heavy chain (V_H_) regions of daratumumab (human IgG1/κ) [34] cloned into the human κ light chain and the human ε (the classic secreted isoform) or human γ1 heavy chain expression vectors [35,36], respectively; all were generously provided by Dr. Sherie L. Morrison (University of California, Los Angeles, CA, USA). Both antibodies were expressed in the murine myeloma cell line Sp2/0-Ag14, expanded in roller bottles, and purified from cell culture supernatants as previously described [23,25,37]. The anti-CD38 IgG1 was purified using Protein A-Sepharose 4B, Fast Flow immunoaffinity matrix (Sigma-Aldrich, St. Louis, MO, USA), and the anti-CD38 IgE was purified using an immunoaffinity column containing anti-human IgE (omalizumab, Xolair^®^, Genentech, Inc. San Francisco, CA, USA) coupled to cyanogen bromide-activated Sepharose^®^ (GE Healthcare, Piscataway, NJ, USA) [23,25]. IgE antibodies were eluted from the column in a manner similar to elution from IgG antibodies from columns containing protein A, as described previously [37]. A human IgE/κ isotype control specific for the hapten dansyl (5-dimethylamino naphthalene-1-sulphonyl chloride) [35] was also produced in murine myeloma cells using the methods described above. All antibodies were quantified using the bicinchoninic acid (BCA) protein assay (Thermo Fisher Scientific). Molecular weight (m.w.) and assembly of purified antibodies were visualized using SDS-PAGE gels under non-reducing and reducing conditions as described for other IgE antibodies [23,25]. Size exclusion chromatography (SEC) was performed in the UCLA-DOE Institute Protein Expression Technology Center. A 500 μL sample (300 μg) of the anti-CD38 IgE was injected on a Superdex 200 10/300 GL size exclusion column (Cytiva Life Sciences, Marlborough, MA, USA) equilibrated with protein buffer (150 mM NaCl, 50 mM Tris-HCl, pH 7.8) using a flow rate of 0.5 mL/min on a Bio-Rad NGC chromatography system (Hercules, CA, USA).

### 2.3. CD38 Antigen and FcεRI Binding Analysis

The specificity of the antibody variable region for human CD38 and binding of the Fc region to human FcεRI were assessed using flow cytometry. 5 × 10^5^ MM.1S cells expressing CD38 or RBL SX-38 expressing human FcεRI were incubated with either vehicle (RPMI + 10% FBS), 2 μg IgE isotype control (non-targeting IgE antibody), anti-CD38 IgG1, or anti-CD38 IgE in 100 μL of RPMI with 10% FBS on ice for 1 h. Cells were washed with RPMI with 10% FBS and incubated for 30 min on ice with a phycoerythrin (PE)-conjugated goat F(ab′)_2_ anti-human κ antibody (Thermo Fischer Scientific) to detect primary antibody binding. Samples (10^4^ events per sample) were analyzed on a BD LSRII analytical flow cytometer (BD Biosciences, San Jose, CA, USA) and histograms were generated using FCS Express 3 (De Novo Software, Los Angeles, CA, USA).

### 2.4. In Vitro Degranulation Assay

This assay, based on β-hexosaminidase release into the culture medium, was performed as previously described [23,25] with some modifications. RBL SX-38 cells (2.5 × 10^5^ cells in 500 μL growth medium per well) were seeded onto a 24-well cell culture plate and incubated overnight at 5% CO_2_ and 37 °C. Then, cells were sensitized with 1 μg of anti-CD38 IgE, anti-CD38 IgG1, IgE isotype control in 500 μL of assay buffer (5 mM KCl, 125 mM NaCl, 20 mM HEPES, 1.5 mM CaCl_2_, 1.5 mM MgCl_2_; pH 7.4), or assay buffer alone for 2 h at 5% CO_2_, 37 °C. The assay buffer was then replaced with either 5 × 10^4^ MM.1S cells in 500 μL of assay buffer or assay buffer alone and incubated for a further 2 h. Cell supernatant from each treatment was plated in triplicate onto a fresh 96-well plate, and substrate [2.5 mM *p*-nitrophenyl-*N*-acetyl-β-D-glucosamine (MilliporeSigma, Burlington, MA, USA) in 50 mM citrate buffer (50 mM citric acid, 50 mM tribasic sodium citrate; pH 4.5)] was added into each well. The plate was then incubated for 1 h at 37 °C. The reaction was terminated by the addition of sodium carbonate buffer (50 mM sodium carbonate, 50 mM sodium bicarbonate; pH 10) and absorbance at 405 nm was measured using a FilterMax F5 multi-mode microplate reader (Molecular Devices, Sunnyvale, CA, USA). β-hexosaminidase release in experimental samples is expressed as a percent of total content within the basophilic cells as determined by separate treatment with 1% Triton X-100 in phosphate-buffered saline (PBS). Statistical analysis was performed using Student’s *t*-test (Microsoft^®^ Excel for Mac Version 16.43, Microsoft Corporation, Redmond, WA, USA).

### 2.5. In Vivo Passive Cutaneous Anaphylaxis (PCA) Assay

Since human IgE is not recognized by murine FcεRI nor FcεRII [22,38,39,40], BALB/c transgenic mice expressing the α subunit of the human FcεRI and responsible for IgE binding [14,23,25,38] (a kind gift from Dr. Jean-Pierre Kinet, Beth Israel Deaconess Medical Center, Boston, MA, USA) were used for this assay. Anaphylaxis is a reaction mediated by FcεRI and the assay was performed as previously described with modifications [14,23,25,38]. Briefly, 0.25 μg of an antibody in 50 μL or vehicle (PBS) control was administered intradermically (i.d.) into the previously shaved skin of transgenic mice. After 1 h, 25 μg of an anti-human κ light chain antibody (Sigma-Aldrich), used to artificially cross-link the human IgE, in 250 μL of 1% Evans blue tetra-sodium salt (Tocris Bioscience, Ellisville, MO, USA) in PBS, was administered intravenously (i.v.) through the tail vein. After 30 min, when visible extravasation of the blue dye into the surrounding tissue was observed, mice were euthanized, and the skin removed. Pictures were analyzed using ImageJ 1.53t (NIH, Bethesda, MD, USA), converting them into 32 bit images, inverted, and displayed as histograms using the histogram command.

### 2.6. ADCC/ADCP Analysis

Monocytes were isolated from human peripheral blood mononuclear cells (PBMCs) obtained from healthy donors (obtained from the UCLA CFAR Virology Core Laboratory, Los Angeles, CA, USA) using the EasySep™ Human Monocyte Isolation Kit (STEMCELL Technologies, Vancouver, BC, Canada). PBMCs were either treated with 10 ng/mL of recombinant human interleukin 4 (IL-4; STEMCELL Technologies) in AIM-V serum-free medium (Thermo Fisher Scientific) with 5% FBS and incubated for 20 h at 5% CO_2_ and 37 °C, or differentiated into macrophages and activated to express an M1 phenotype using ImmunoCult™-SF Macrophage Medium (STEMCELL Technologies) following the manufacturer’s instructions. Flow cytometric analysis of ADCC and ADCP was performed as described [41] with modifications. Briefly, target cells (MM.1S) were labeled with carboxyfluorescein succinimidyl ester (CFSE, Thermo Fisher Scientific) at 250 ng per 10^6^ cells per mL for 10 min as per the manufacturer’s instructions. Target cells were then washed and incubated in fresh growth medium at 5% CO_2_ and 37 °C overnight. The next day effector cells (IL-4 treated monocytes or M1 macrophages) and target cells were mixed together at a 5:1 effector to target ratio and incubated with 5 μg/mL of antibody in growth medium for 2.5 h at 5% CO_2_ and 37 °C in quadruplicate. Cells were then washed in growth medium and incubated with a PE-conjugated mouse anti-human CD89 antibody (BD Biosciences) for 25 min to identify phagocytes. DAPI (4′,6-diamidino-2-phenylindole) was added at a 1 μg/mL concentration to identify dead cells with cells treated with saponin used as a positive control for cell lysis. Samples were analyzed on a BD LSRII analytical flow cytometer, and 5 × 10^4^ events were collected. CFSE^+^/PE^+^ cells designate ADCP events and CFSE^+^/DAPI^+^ cells designate ADCC events. Statistical analysis was performed using Student’s *t*-test (Microsoft^®^ Excel for Mac Version 16.43, Microsoft Corporation).

### 2.7. In Vivo Antitumor Activity

C.B-17 SCID-Beige female mice (8–12 weeks old), obtained and housed in the Defined Flora Mouse Facility in the Department of Radiation Oncology at UCLA, were exposed to whole-body sublethal irradiation of 3 Gy (GammaCell40 irradiator ^137^Cs, Best Theratronics, Ltd., Ottawa, ON, Canada) a day before xenograft implantation, and were implanted with 5 × 10^6^ MM.1S human MM cells i.v. via the tail vein. On Day 1 and Day 7 post-implant, mice were treated i.v. with buffer (PBS) control, 100 μg anti-CD38 IgE, 5 × 10^6^ PBMCs (as a source of monocyte effector cells), or 5 × 10^6^ PBMCs combined with 100 μg anti-CD38 IgE. For the last treatment group, the IgE and PBMCs were added to the same tube and placed on ice until injections could be performed. PBMCs were obtained from healthy donors (UCLA CFAR Virology Core Laboratory). The PBMCs for each independent study were from separate donors. Mice were then observed for the onset of hind-limb paralysis (end point), and the number of days survived was recorded. Kaplan–Meier survival analysis was used to analyze the data and different Kaplan–Meier survival plots were compared using the log-rank test in GraphPad Prism, Version 9 (GraphPad Software, Inc., La Jolla, CA, USA).

## 3. Results

### 3.1. The Anti-CD38 IgE Is Properly Assembled and Secreted

A non-reduced SDS-PAGE (Figure 1A) confirmed that the anti-CD38 IgG1 and anti-CD38 IgE have m.w. that are consistent with the m.w. of human IgG1 (146 kDa) and human IgE (188 kDa) [14]. Under reducing conditions, where disulfide bonds are cleaved and thereby allowing for the separation of the light and heavy chains by SDS-PAGE (Figure 1A), the observed m.w. of κ light, γ and ε heavy chains are also consistent with their m.w. [14]. The ε heavy chain is larger than the γ heavy chain due to the presence of an additional constant domain on the ε heavy chain (Figure 1B) [14].

SEC analysis of the anti-CD38 IgE was conducted to determine if any aggregates were present (Appendix A). This analysis and comparison with a standard curve indicate the retention volume of the peak containing the IgE was consistent with a non-aggregated, monomeric form of the anti-CD38 IgE.

### 3.2. The Anti-CD38 IgE Binds Both CD38 and FcεRI, Induces Degranulation In Vitro and In Vivo, and Is Able to Elicit IgE Fc-Mediated Effector Functions

To further explore the properties of the anti-CD38 IgE, we first assessed the binding of the antibody to the antigen and to FcεRI. Both the anti-CD38 IgE and IgG1 antibodies showed binding to MM.1S cells, which express CD38 (Figure 2). The binding profiles of the two antibodies are similar, as expected since they both contain the same variable regions and thus bind the same epitope of CD38. The anti-CD38 IgE antibody binds RBL SX-38 cells that express human FcεRI (Figure 2). The IgE isotype control antibody also binds these cells, while the anti-CD38 IgG1 does not. These results are expected since binding to RBL SX-38 cells occurs through the Fc region of the antibody. Since the binding properties of the anti-CD38 IgE are as expected, we next explored the functional activity of the antibody. In the presence of MM.1S cancer cells expressing CD38, the anti-CD38 IgE induces the degranulation of RBL SX-38 cells in vitro (Figure 3). As expected, incubation of the human MM.1S cell line with the anti-CD38 IgG1 counterpart did not trigger degranulation, showing only basal level signal since degranulation is mediated by antibodies of the IgE class. The anti-CD38 IgE, but not the anti-CD38 IgG1, also induces local (cutaneous) anaphylaxis (type I hypersensitivity) in human FcεRIα transgenic mice as a result of the degranulation of skin mast cells loaded with the IgE antibody and artificially cross-linked with an anti-human κ antibody (Figure 4). This reaction is demonstrated by the visual extravasation of the blue dye into the surrounding skin, which is quantified and further demonstrated by the histogram underneath the skin pictures (Figure 4). To further explore the functional activity of the anti-CD38 IgE, we assessed additional antibody-mediated effector functions. The anti-CD38 IgE mediates ADCP in monocytes incubated with IL-4 (Figure 5A) and ADCC and ADCP in monocyte-derived M1 macrophages (Figure 5B). Taken together, these studies demonstrate that the anti-CD38 IgE is a functional IgE antibody and mediates antitumor properties in vitro.

### 3.3. Anti-CD38 IgE Prolongs Survival in an In Vivo Model of MM in the Presence of Human PBMCs

We then sought to evaluate the potential antitumor effects of the anti-CD38 IgE antibody in a disseminated mouse model of MM. SCID-Beige mice treated with anti-CD38 IgE and human PBMCs as a source of monocytes (effector cells), survive significantly longer (Figure 6, median survival = 41 days) than either IgE alone (32 days, *p* = 0.0004), PBMCs alone (37 days, *p* = 0.0113), or buffer (32 days, *p* = 0.0153). These results also show no difference in the survival of anti-CD38 IgE-treated mice versus buffer treated mice (*p* = 0.3343) and a slight, but significant, increase in survival in PBMC-only treated mice compared to buffer treated mice (*p* = 0.0457). The observation that mice given PBMCs alone survived longer than buffer controls is consistent with previous studies [26,27]. These results were replicated in a second experiment in which mice were treated with the anti-CD38 IgE and PBMCs on Days 1 and 9 (Appendix A).

## 4. Discussion

CD38 is a multifunctional type II transmembrane glycoprotein that acts as a receptor, an adhesion molecule interacting with CD31, and as an ectoenzyme (ADP-ribosyl cyclase/cyclic ADP-ribose hydrolase) [42,43,44]. Expression of CD38 has been found in chronic lymphocytic leukemia (CLL), acute lymphoblastic leukemia (ALL), acute myeloid leukemia (AML), aggressive natural killer (NK) cell leukemia (ANKL), NK/T-cell lymphoma, mantle cell lymphoma (MCL), and Waldenström’s macroglobulinemia (WM) [42,43]. CD38 shows especially high and uniform expression on MM cells, in which it plays a relevant role in cancer cell pathology, making it a target of choice for therapeutic antibodies targeting cell surface molecules in MM [42,43,44]. In this light, IgG1 class antibodies targeting CD38, such as daratumumab (Darzalex^®^) and isatuximab (Sarclisa^®^), have benefited MM patients, especially when they are used in combination with other drugs [2]. Unfortunately, in spite of the relevant improvement in patient survival, MM remains incurable [1,2], and thus additional therapeutic interventions are urgently needed.

The success of the IgG1 antibodies daratumumab (Darzalex^®^) and isatuximab (Sarclisa^®^) in MM has ignited enthusiasm to develop additional CD38-targeting agents. To take advantage of the unique properties of IgE antibodies as cancer therapeutics and to address unmet medical needs in MM, we now report, for the first time, the development of a new fully human anti-CD38 IgE as a potential targeted immunotherapy for MM. The anti-CD38 IgE is properly assembled and secreted and exhibits the correct m.w. as demonstrated by SDS-PAGE and binds antigen (CD38) and FcεRI as demonstrated by flow cytometry. We also showed that anti-CD38 IgE triggers in vitro degranulation of rat basophilic leukemia cells expressing human FcεRI (RBL SX-38) in the presence of the human MM cells expressing CD38 (MM.1S). This result is relevant since MM tumors are infiltrated by mast cells where they contribute to angiogenesis and MM growth [45,46], but in the presence of a tumor-specific IgE these mast cells would degranulate resulting in an acute inflammatory immune response as well as the release of pro-apoptotic compounds and stimulators of the immune response [29,47,48,49], which may potentially result in antitumor activity.

Relevant Fc effector functions of human IgE are ADCC and ADCP elicited by monocyte/macrophages resulting in the induction of cancer cell death [14,21,22,50,51]. To assess ADCC and ADCP, we used human monocytes, circulating precursors for tissue macrophages, as well as human M1 macrophages as effector cells. It is important to mention that these effector cells express both FcεRI and FcεRII [13,14,21,22]. The induction of ADCC and ADCP was assessed by three-color flow cytometry, and we found that in the presence of MM.1S cells as targets and monocytes or M1 macrophages as effectors, the anti-CD38 IgE antibody increases the level of ADCP and ADCC/ADCP, respectively. This result is also important since macrophages are major components of the MM stroma and may also provide support to the malignant plasma cells protecting them from drug-induced apoptosis [52,53]. However, macrophages are potent ADCC and ADCP effectors when tumors are targeted by IgE, an antibody class also capable of re-programing macrophages to fight the tumor [13,14,50,51]. Thus, our results suggest that monocytes and macrophages armed with the tumor-targeting anti-CD38 IgE antibody would trigger antitumor activity in vivo. Importantly, the use of MM.1S cells in our studies is relevant given the higher MM incidence and mortality of individuals of African ancestry compared other races [2,54].

The anti-CD38 IgE induces local (cutaneous) anaphylaxis in transgenic mice expressing the human FcεRIα subunit that binds IgE [14,38], triggered by artificially cross-linking the IgE antibody on the surface of skin mast cells as demonstrated in the PCA assay. This local anaphylactic reaction is not dependent on the CD38 specificity of the antibody, but this result shows that the IgE antibody is functional and has the capacity to bind FcεRI and induce anaphylaxis in vivo. The release of the dye (Evans blue) occurs due the ability of IgE to induce a local increase in vascular permeability, a “gatekeeper” IgE effect that can also be exploited to facilitate tumor targeting of other therapeutic agents that would better penetrate the tumor microenvironment due the increase in the tumor blood vessel permeability [21]. This is per se another application of IgE in combination therapy, which may also have diagnostic value. The mouse model expressing the “humanized” FcεRI used in the PCA assay is required since human IgE is not recognized by murine FcεRI [22,38,40]. However, since this mouse model is immunocompetent, it cannot be used to study the antitumor activity against human MM tumors. Another drawback is the lack of available syngeneic murine MM cell lines expressing human CD38 that are capable of growing in immunocompetent mice. Additionally, the anti-CD38 IgE, similar to daratumumab, does not cross-react with murine CD38 since they have the same variable regions [55,56]. To overcome these limitations, we used a xenograft mouse model (SCID-Beige) reconstituted with PBMCs as a source of monocytes as IgE effector cells [21,26,38]. We found that the treatment of mice with anti-CD38 IgE and PBMCs prolongs the survival of mice bearing disseminated MM.1S tumors compared to all other treatments. These in vivo results are consistent with the in vitro results using monocytes/macrophages as effector cells. The efficacy of anti-CD38 IgE in SCID-Beige mice reconstituted with human PBMCs is particularly relevant given the limitation in type and number of IgE effector cells (only monocytes) administered into the mice and the fact that PBMCs are short lived. These mice also lack adaptive immunity since they do not have mature B and T cells due to the SCID mutation [38,57] precluding the assessment of the immunostimulant properties of IgE. Importantly, IgE is capable of enhancing antigen uptake and presentation in antigen-presenting cells such as dendritic cells, boosting the magnitude and duration of the humoral and cellular adaptive immune response, and also linking the innate with the adaptive immune response [23,25,58,59,60]. Thus, the antitumor activity of the anti-CD38 IgE antibody is expected to be stronger in the presence of a complete immune cell repertoire as would occur in humans.

A major concern in the use of IgE as a cancer therapeutic is the possible induction of a systemic type I hypersensitivity (anaphylactic) reaction, which unfortunately, cannot be addressed in the animal models used in these studies. This is a concern since circulating soluble CD38 (sCD38) has been detected in MM patients [42,61]. However, sCD38 is only found in a subset of patients [42,61] and generally at low concentrations [62]. However, since a mono-epitopic interaction of the daratumumab variable regions with sCD38 has been demonstrated [63], a systemic type I hypersensitivity reaction is not expected to occur, as we have described in case of soluble tumor antigens PSA and the extracellular domain of HER2/*neu* (ECD^HER2^) bound to anti-HER2/*neu* IgE and anti-PSA IgE, respectively [23,25]. Although CD38 is highly and uniformly expressed on MM cells, and at relatively low levels on normal hematopoietic cells including myeloid and lymphoid cells, as well as in some cells of non-hematopoietic origin [42,43], CD38 expression on normal hematopoietic cells plasma cells, thymocytes, lymphocytes, activated T cells and B cells, dendritic cells, NK cells, erythrocytes, and platelets, is a concern. However, since CD38 expression on MM cells is higher compared to most normal cells, CD38 is considered to be a tumor-associated antigen [42,43,44,64], combined with the fact that the MM is infiltrated by IgE effector cells including mast cells [45,46,52,53], suggests that the type I hypersensitivity reaction would be preferentially localized in the tumor microenvironment. Potential unwanted side effects, such as systemic anaphylaxis, may also be dose dependent. Further studies are needed to address the potential toxicity of the anti-CD38 IgE. In addition, anaphylactic side effects may also be ameliorated through identification of potentially susceptible patients via skin prick tests and basophil activation test pre-treatment described in the MOv18 IgE Phase I clinical trial [15,30,31]. Moreover, this side effect may also be potentially prevented by the prophylactic administration of anti-histamine drugs, such as diphenhydramine (Benadryl), before the IgE therapy, which is recommended for other FDA-approved monoclonal antibodies such as daratumumab (Darzalex^®^) [65,66], without affecting the antitumor benefits of the IgE therapy.

## 5. Conclusions

In summary, our studies describe a fully human anti-CD38 IgE antibody that demonstrates IgE activity in vitro and in vivo including antitumor activity against human MM. To our knowledge, this is the first IgE to show both in vitro and in vivo efficacy against a hematological cancer, more specifically a plasma cell cancer (in an aggressive disseminated murine model) and the first to target CD38. Taken together, our results suggest that the anti-CD38 IgE antibody would be effective in humans affected with MM and potentially in other hematopoietic malignancies expressing CD38. However, further studies are necessary to assess the suitability of the anti-CD38 IgE antibody as a MM therapy. Regardless of whether or not anti-CD38 IgE becomes a MM therapeutic, the present studies support the use of IgE-based therapy targeting different antigens for the treatment of MM and potentially other hematopoietic malignancies. It is also important to point out that in cancer therapy, and especially in MM, combinatorial treatment with multiple drugs is part of the standard of care. In this light, the IgE technology is not necessarily a replacement or competitor of other therapeutic approaches, including antibodies of the IgG class, but can be used in new combination cancer therapy strategies.

## 6. Patents

The following patent application has been filed: No. PCT/US2021/048714 (M.L.P., P.V.C., M.N., and T.R.D.-W.).

## Figures and Tables

**Figure 1 cancers-15-04533-f001:**
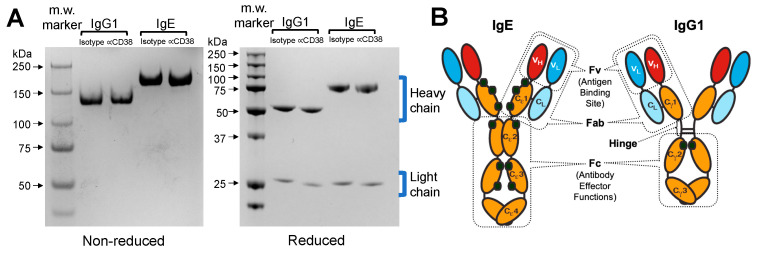
SDS-PAGE analysis. (**A**) Purified isotype and anti-CD38 IgG1 and IgE (2 μg) were electrophorized under non-reducing and reducing conditions. The positions of m.w. markers are indicated at the left. (**B**) Diagrammatic representation of the structure of an IgE and an IgG1 antibody. Antibodies are composed of two pairs of identical heavy (H) and light (L) chain proteins linked by disulfide bonds forming H_2_L_2_ heterotetramers. The Fab region consists of a constant and a variable domain from each heavy and light chain. The variable region (Fv), composed of both the heavy and light chain variable domains, is responsible for antigen binding and is located at the amino-terminus of the antibody. The remaining constant regions include the Fc portion of the antibody and are responsible for the effector functions. Black circles denote *N*-linked glycosylation sites. The hinge region, which provides flexibility, joins the Cγ1 and Cγ2 domains in IgG1. Cε2 replaces the hinge region in IgE. The IgG1 has a m.w. of 146 kDa while IgE has a m.w. of 188 kDa. The figure in Panel (**B**) was reprinted from Figure 7.2 of Daniels et al., 2010 (Ref. [21]), with kind permission from Springer Nature.

**Figure 2 cancers-15-04533-f002:**
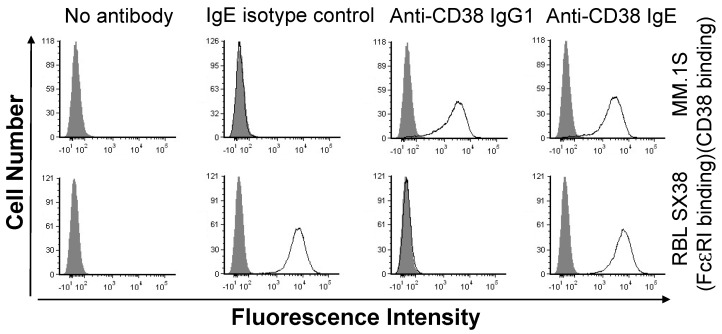
Binding to antigen and FcεRI analyzed by flow cytometry. MM.1S cells expressing CD38 or RBL SX-38 expressing human FcεRI were incubated with either buffer (diluent) control, 2 μg IgE isotype control, anti-CD38 IgG1, or anti-CD38 IgE, in 100 μL of RPMI + 10% FBS on ice for 1 h. Cells were washed and antibody binding was detected by incubating with PE-conjugated goat F(ab′)_2_ anti-human κ antibody for 30 min on ice. Grey-filled histograms represent cells incubated with the PE-conjugated secondary antibody only (“no antibody”). Empty black line histograms represent cells incubated with primary antibodies (IgE isotype control, anti-CD38 IgG1, or anti-CD38 IgE) prior to staining with the PE-conjugated secondary antibody. The results are representative of two independent experiments.

**Figure 3 cancers-15-04533-f003:**
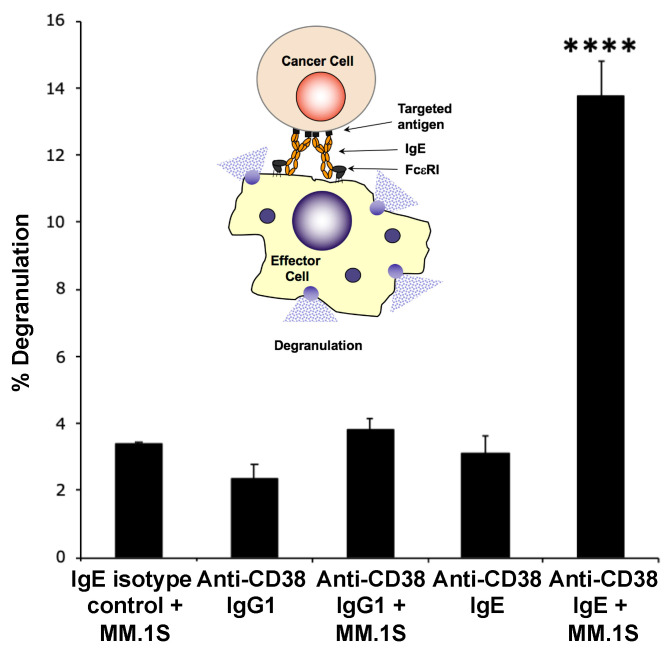
In vitro degranulation assay. RBL SX-38 cells were sensitized with 1 μg of either anti-CD38 IgG1, anti-CD38 IgE, or isotype IgE control for 2 h. Supernatant was then replaced with either buffer or MM.1S cells. Release of β-hexosaminidase in the supernatant was measured enzymatically. The mean and standard deviation of triplicate samples are shown. **** *p* < 0.0001 (Student’s *t*-test) compared to each control group. The inlet figure is a diagrammatic representation of how cancer cell-IgE-mediated cross-linking of FcεRI on effector cells (basophils or mast cells) leads to degranulation. The results are representative of two independent experiments (data from a replicate study are shown in Appendix A). The inlet figure was reprinted from Figure 2 (Panel A) of Leoh et al., 2015 (Ref. [14]), with kind permission from Springer Nature.

**Figure 4 cancers-15-04533-f004:**
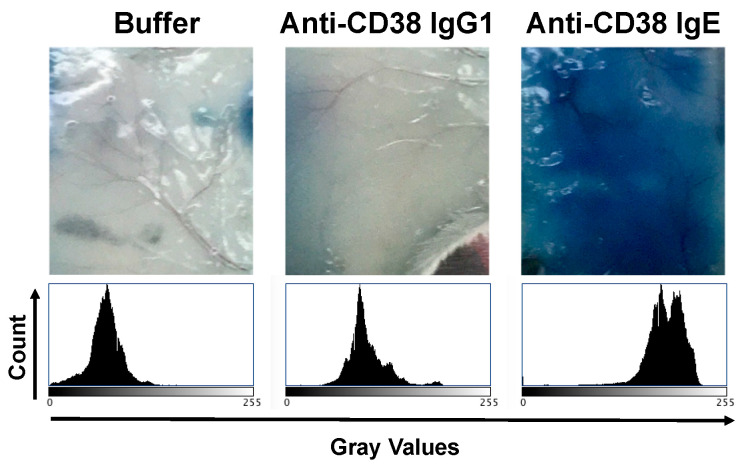
In vivo PCA assay in human FcεRIα transgenic mice. Images are of the skin of mice administered i.d. with 50 μL of 5 μg/mL of anti-CD38 IgG1, anti-CD38 IgE, or buffer (PBS) control. After 1 h, 250 μL of 1% Evans blue dye in PBS with 25 μg of anti-human κ light chain antibody was administered i.v., for cross-linking of the IgE-FcεRI complex to trigger mast cell degranulation. The mice were euthanized after 30 min. Cutaneous anaphylaxis was assessed visually by the blue dye extravasation from blood vessels into the skin due to vasodilation. Dye was quantified by ImageJ 1.53t. The results are representative of two independent experiments.

**Figure 5 cancers-15-04533-f005:**
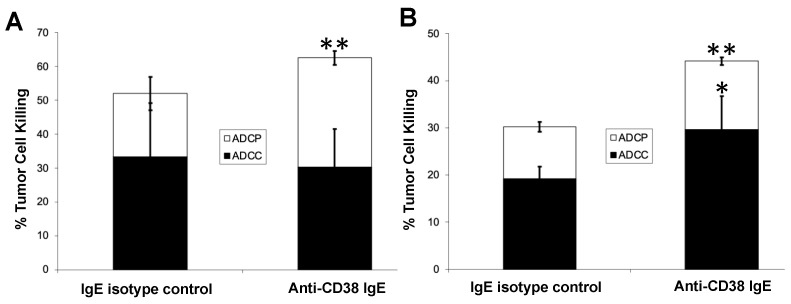
ADCC/ADCP assessed by three-color flow cytometry. Monocytes are isolated and either (**A**) treated with 10 ng/mL of IL-4 for 20 h or (**B**) differentiated into macrophages and activated towards an M1 phenotype, then used as effector cells against CFSE labeled MM.1S target cells treated with either IgE isotype control or anti-CD38 IgE antibody. After incubating effector and target cells at 5:1 effector to target ratio for 2.5 h, cells were stained with both a PE-conjugated mouse anti-human CD89 antibody and DAPI, then analyzed by flow cytometry (5 × 10^4^ events were collected). ADCP was defined as CD89-PE^+^ and CFSE^+^ events while ADCC was defined as CFSE^+^ and DAPI^+^ events. Groups were compared using Student’s *t*-test (* *p* < 0.05, ** *p* < 0.01). The results are representative of two independent experiments (data from a replicate study are shown in Appendix A using PBMCs from a different donor).

**Figure 6 cancers-15-04533-f006:**
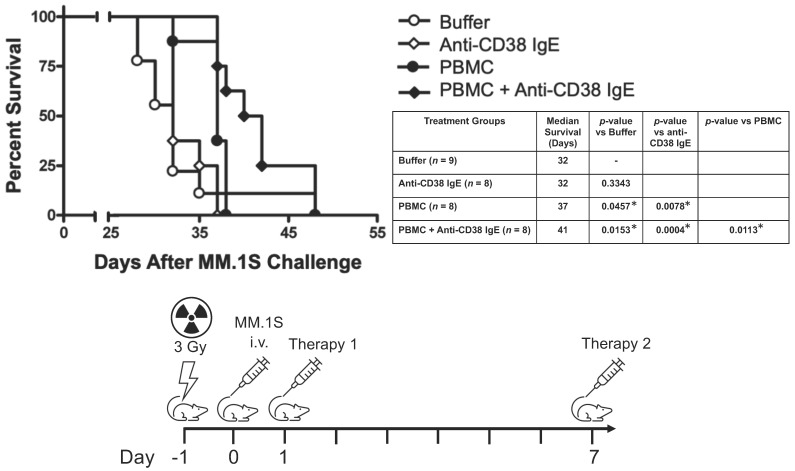
In vivo antitumor activity. Kaplan-Meier survival plot analysis showing survival of SCID-Beige mice challenged with 5 × 10^6^ MM.1S cells i.v. via the tail vein to create a model of disseminated disease. On Day 1 and Day 7 after tumor challenge, mice were treated i.v. with buffer (PBS) control (*n* = 9), 100 μg of anti-CD38 IgE (*n* = 8), 5 × 10^6^ PBMCs (*n* = 8), or 100 μg of anti-CD38 IgE and 5 × 10^6^ PBMCs (*n* = 8) in 250 μL. Mice were then observed for the onset of hind-limb paralysis (end point) and survival recorded. The difference in survival among the different groups was analyzed using the log-rank test and shown in the table (* *p* < 0.05). The results are representative of two independent experiments (data from a replicate study are shown in Appendix A).

## Data Availability

The dataset generated during and/or analyzed during the current study is available from the corresponding author (M.L.P.) upon reasonable request.

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
