# Peer review of "A Fully Human IgE Specific for CD38 as a Potential Therapy for Multiple Myeloma"

_cancers, 2023, doi:10.3390/cancers15184533_

Round 1

Reviewer 1 Report

This is a very interesting manuscript that describes a substantially novel antibody approach to potentially treat CD38-positive tumors, such as multiple myeloma. A fully human anti-CD38 antibody of the IgE type has been synthesized and thoroughly characterized by the authors. The experiments were well-designed, state-of-the-art, and were well-done. The results convince the readers that the antibody has potential. Unfortunately a high toxic potential, as well. Due to the lack of proper transgenic mouse strains, toxicity analysis could not be done at this stage of development, this awaits possible Phase I human trials.

I discovered only one error in the reasoning of the authors. In line 414 they indicate that soluble CD38 is not present in most myeloma patients, that is not necessarily correct. Most active myeloma patients have some soluble CD38 present in their sera, the amount is somewhat proportional to the tumor burden. The same could be said about circulating CD38-positive tumorous and non-tumorous cells.  Airway CD38-expression is also of major concern. Nevertheless, possibly a "clearing" daratumumab pretreatment dose could be used to eliminate this high-risk problem of anaphylaxis.

In experiment 3.3, a control with daratumumab could also increase the validity of the conclusions.

Line 58, the other approved anti-CD38 antibody, isatuximab should also be mentioned in this paragraph.

I see no problem here

Author Response

We thank Reviewer 1 for their thoughtful and insightful comments. Below is our response to their feedback as well as the reviewer comment that it addresses directly preceding our response.

Comment 1: I discovered only one error in the reasoning of the authors. In line 414 they indicate that soluble CD38 is not present in most myeloma patients, that is not necessarily correct. Most active myeloma patients have some soluble CD38 present in their sera, the amount is somewhat proportional to the tumor burden. The same could be said about circulating CD38-positive tumorous and non-tumorous cells. Airway CD38-expression is also of major concern. Nevertheless, possibly a "clearing" daratumumab pretreatment dose could be used to eliminate this high-risk problem of anaphylaxis.

Authors’ response: We thank the reviewer for this information. We have found limited information on the level of soluble CD38 in the circulation of multiple myeloma (MM) patients. Nijhof et al. 2016 (DOI: 10.1182/blood-2016-03-703439) states “Soluble CD38, which may also bind daratumumab, was evaluated in 110 of the 148 patients and detected in only 2 cases.” Although it is only in a small survey of patients, from this we inferred that soluble CD38 is either uncommon or expressed in low enough levels to be undetectable most times in (MM patients. Another article by Li et al. 2018 (DOI: org/10.1016/j.aca.2018.04.061) states that the levels of sCD38 in MM patients is extremely low due to dilution in the circulation system (<1.5 ng/mL). This article showed increased levels in MM patients compared to normal donors using an ultra-sensitive assay developed by this group. They also showed soluble CD38 levels correlated with disease progression. However, this article did not show that most MM patients have circulating soluble CD38. It is present only in a subset of MM patients. We welcome the reviewer’s correction and ask for any additional references that we may cite to show a greater prevalence of soluble CD38 in MM patients. As we were made aware of the possible greater prevalence of soluble CD38 in myeloma patients, we opted to change the line (458 - 462) from “However, soluble CD38 is only found in a small number of patients” to “However, soluble CD38 is only found in a subset of patients, is extremely low due to dilution in the circulatory system (<1.5 ng/mL). Soluble CD38 is increased in MM patients compared to normal donors and is correlated with disease progression as detected by an ultra-sensitive nanobody based assay. ” and have added this additional reference (Li et al. 2018).  Furthermore, as described in the manuscript (lines 462-466), the interaction of the anti-CD38 IgE with the antigen is mono-epitopic, meaning that the antibody will only bind two soluble CD38 molecules (one for each variable region). Thus, the interaction with soluble CD38 is not expected to induce degranulation of effector cells in the circulation, independent of how much soluble CD38 is present. 

Potential systemic anaphylaxis is still a concern, as we mentioned in the manuscript. However, the focus of this manuscript was to show that the antibody has antitumor activity in vitro and in an animal model. Of course further studies need to focus on the potential toxicities including the induction of systemic anaphylaxis. Whether or not expression of CD38 on the surface of air way cells is high enough to trigger degranulation/anaphylaxis remains to be determined. We thank the reviewer for bringing this to our attention. It is also unknown if the level of circulating malignant cells is high enough to trigger this reaction. MM cells are typically in the bone marrow so the amount of cells in the circulation is expected to be low, except in the case of plasma cell leukemia (for which the anti-CD38 IgE may not be a good treatment option).  The “clearing” daratumumab pretreatment dose is an intriguing idea and we thank the reviewer for this suggestion. However, daratumumab and our anti-CD38 IgE bind to the same epitope on CD38, which could be problematic. Perhaps careful timing of treatments or using a different anti-CD38 IgG antibody, such as isatuximab since they have distinct epitope regions (DOI: 10.3390/cells8121522), may be a more effective option. There are also other ways to ameliorate anaphylaxis if necessary. Exclusion criteria can also be explored to further circumvent any potential toxicities. Additionally, it is well known that antigen-IgE immune complexes can enhance antigen presentation and lead to the induction of a secondary immune response that could add to the anti-cancer effect of the IgE. Since so much is unknown at this time, we prefer not to speculate too much on this subject in the current manuscript, but we agree with the reviewer that this is an important point that must be addressed in future studies.

Comment 2: In experiment 3.3, a control with daratumumab could also increase the validity of the conclusions.

Authors’ response: In principle we agree with the reviewer that such a comparison would be academically interesting. However, our interest in developing IgE antibodies is not to show superiority to their IgG1 counterparts, but to develop alternative strategies to treat MM. The two antibodies are not mutually exclusive and may be used in careful combinatorial treatment regimen.

Additionally, we know that the mouse models for evaluating human IgE antibodies are challenging. Therefore, a direct comparison between the anti-CD38 IgG1 antibody and the anti-CD38 IgE variant is difficult. In the xenograft mouse model used in our study, the human PBMC is the sole source of IgE effector cells since human IgE is unable to bind mouse FceR. In contrast, human IgG1 binds to both human and mouse FcgR, and thus would have a wider panoply of effectors to draw upon in the model. Thus, daratumumab would have an advantage and the results may be misleading. Such a comparison would not be meaningful in our mouse model.

Comment 3: Line 58, the other approved anti-CD38 antibody, isatuximab should also be mentioned in this paragraph.

Authors’ response: We thank the reviewer for the suggestion and have added a sentence to the introduction.

Reviewer 2 Report

In this study, Candelaria et el. developed a novel anti-CD38 IgE and tested its potential anti-myeloma activity by using different in vitro and in vivo assays.

The manuscript is interesting and innovative and may provide the basis for the introduction of anti-CD38 IgE moAbs in clinical practice. Nevertheless, some points should be revised.

Major points:

-        Authors should compare the % of tumor cell killing of the anti-CD38 IgE to the killing activity of an anti-CD38 IgG moAb (at least in the in vitro studies).

-        As the activation of monocytes/macrophage may induce the release of inflammatory cytokines (e.g INF-gamma, or TNFalpha) that, in turn, may activate an anti-tumor immune response by stimulating other immune cells, I suggest to evaluate the release of this inflammatory cytokines to dissect the effect of the anti-CD38 IgE MoAb on the immune microenvironment.

-        NK cells are mainly involved in the anti-tumor activity and in the killing of tumor cells. Although NK cells do not express the FcƐRI/ FcƐRII, are the anti-CD38 IgE MoAb able to indirectly stimulate NK cells, for instance through the release of pro-inflammatory cytokines? The expression of NK cells activation marker (e.g. CD137, CD69, CD107a) should be evaluated.

Minor point:

-        The paragraph 3.2 of the results is the most important of the manuscript. The authors should better describe the figures (e.g. Figure 4 and Figure 5) to support their conclusions.

 Minor editing of English language required

Author Response

We thank Reviewer 2 for their thorough and helpful comments. Below is our response to their feedback as well as the reviewer comment that it addresses directly preceding our response.

Major point 1: Authors should compare the % of tumor cell killing of the anti-CD38 IgE to the killing activity of an anti-CD38 IgG moAb (at least in the in vitro studies).

Authors’ response: Daratumumab is already known to mediate ADCC, ADCP, and CDC as cited in the manuscript. Although to compare the effector functions of the anti-CD38 IgG1 antibody with the anti-CD38 IgE variant would be interesting, such a comparison would be of limited informative value since the effector cells and FcR expression levels and mechanism of engagement of these receptors is different between IgG and IgE antibodies. Moreover, in this study we are using just one type of effector cells, but different results may be obtained with different effector cells. Additionally, it is not our goal to state that the IgE has more antitumor activity compared to the IgG1, but that it is an alternative therapy that activates different immune cells and thus, has value for the treatment of MM. The two antibodies are not mutually exclusive and may even be used together as part of a combination therapy. Further studies are needed to address this possibility.

Major point 2: As the activation of monocytes/macrophage may induce the release of inflammatory cytokines (e.g INF-gamma, or TNFalpha) that, in turn, may activate an anti-tumor immune response by stimulating other immune cells, I suggest to evaluate the release of this inflammatory cytokines to dissect the effect of the anti-CD38 IgE MoAb on the immune microenvironment.

Authors’ response: We thank the reviewer for interesting and worthwhile suggestions that would further elucidate the mechanism of IgE anti-tumor therapy. We expect this to be the case. However, such studies would take considerable time to carefully and fully interrogate since this is a complex phenomenon with many factors involved. We shall incorporate the reviewer’s suggestions in our continued studies of IgE, not only on monocytes and macrophages, but also on basophils, eosinophils, and mast cells.

Major point 3: NK cells are mainly involved in the anti-tumor activity and in the killing of tumor cells. Although NK cells do not express the FcƐRI/ FcƐRII, are the anti-CD38 IgE MoAb able to indirectly stimulate NK cells, for instance through the release of pro- inflammatory cytokines? The expression of NK cells activation marker (e.g. CD137, CD69, CD107a) should be evaluated.

Authors’ response: Yes, this is an excellent point that will need to be elucidated in a model that has functional NK cells and in human clinical trials. The xenograft model used in our study (SCID-Beige mice) are needed for the MM cells to engraft. However, since these mice have functionally impaired NK cells, it is not the ideal model system to interrogate this question.

Minor point 1: The paragraph 3.2 of the results is the most important of the manuscript. The authors should better describe the figures (e.g. Figure 4 and Figure 5) to support their conclusions.

Authors’ response: We agree with the reviewer that this is a key paragraph. We thank the reviewer for pointing out its shortcomings. Additional sentences describing the figures were added to more fully describe the studies and support the conclusions, as suggested.

Reviewer 3 Report

Interesting and scientifically sound work in an attempt to use new therapeutic targets and develop new agents in the treatment of MM. The  data presented are interesting but as expected many further studies are required before any safe conclusions can be drawn about the clinical application of this monoclonal antibody and IgE as a therapeutic target. 

Author Response

We thank Reviewer 3 for their kind comments. However, we are perplexed as in their feedback they indicated that the introduction and background must be improved but did not specify how, or what to include to improve this section. We added to the Introduction a recently published article that is highly relevant to the topic of IgE antibodies for cancer therapy. Additionally, in our responses to reviewer 1 and 2, we have added to the Introduction and the Results section to clarify their comments. We hope this contributes in some way towards allaying this reviewer’s concerns regarding any possible insufficiencies regarding the manuscript background and references.